# Florentine Normative Values for Physical Fitness in Adolescents Aged 14–15 Years

**DOI:** 10.3390/healthcare10122486

**Published:** 2022-12-08

**Authors:** Gabriele Mascherini, Niccolò Buglione, Virginia Ciani, Franco Tirinnanzi, Vittorio Bini, Matteo Levi Micheli

**Affiliations:** 1Exercise Science Laboratory Applied to Medicine “Mario Marella”, Department of Experimental and Clinical Medicine, University of Florence, 50134 Florence, Italy; 2Asics Firenze Marathon, Viale Manfredo Fanti, 2, 50137 Florence, Italy; 3Dipartimento di Medicina e Chirurgia, Università di Perugia, Piazzale Gambuli 1, 06129 Perugia, Italy

**Keywords:** aerobic fitness, percentile physical fitness, reference values, flexibility, strength

## Abstract

Physical fitness during youth is linked to the health of today’s young people and future adults. Therefore, fitness evaluation can direct any physical exercise interventions and verify improvements. The study aims to provide normative values of the physical fitness of the Florentine adolescent population. This cross-sectional study enrolled 1915 subjects aged 14 and 15 from the first and second high school classes. Tests were performed to assess cardiovascular endurance, upper and lower limb strength, running speed and agility, and lower limb muscle flexibility. Sex and age-related percentiles were elaborated. The study included 1028 subjects in the first class (aged 14.3 ± 0.8 years; 54.5% females) and 887 in the second class (aged 15.3 ± 0.7 years; 53.4% females). Males showed higher values than females except for flexibility. Furthermore, both genders show increased results in transitioning from the first to the second class, except for muscle flexibility and running speed. The percentiles of the present study provide normative values of the physical fitness of the 14–15-year-old Florentine adolescent population. These results, characterized by a sociocultural specificity linked to the territory from which they were collected, can contribute to the assessment and improvement of the physical status of young people, promoting current and future health.

## 1. Introduction

In recent decades, notable changes have been described as a centuries-old reduction in young peoples’ physical fitness development [1]. Physical fitness measures the body’s ability to perform physical activity and provides a summative indicator of health [2]. Direct associations have been reported in adolescents between physical fitness and the risk of cardio metabolic disease, adiposity, and mental and cognitive health [3,4]. There is also direct evidence indicating that low levels of physical fitness in adolescence are significantly associated with all-cause mortality later in life [5]. Therefore, it is increasingly important to track fitness levels during youth to identify adverse trends to promote primary prevention or implement early interventions [6].

In this context, physical fitness tests can perform an educational function to improve adolescent physical literacy [7], identify young people with physical and motor development problems, and start planning individual adapted learning objectives [8]. However, to carry out individualized interventions, an evaluation is necessary that provides specific reference values for sex and age, against which the individual and population fitness status can be compared. This allows for an adequate interpretation of adolescent development, providing information to teachers, parents, adolescents, political decision-makers, and young people.

Currently, there are reference values for the physical fitness of children and adolescents, elaborating percentiles by administering specific tests [9,10,11]. Most of these reference values and curves are linked to national and international projects describing large-scale values. Therefore, on the one hand, they show solidity for the sample size; on the other, they could show a loss of specificity in the sociocultural characteristics of subpopulations at places to which one wants to refer [12]. For example, in Italy, reference physical fitness standards for children between the ages of 6 and 11 elaborated on physical tests in northern Italy [13]. However, Italian adolescents have no regulatory benchmarks for physical fitness. This study hypothesizes that physical fitness values increase with age, and that there is an increase in performance in males as early as adolescence.

Therefore, the objective of this study was to obtain and describe the sex and age-specific fitness normative values for the 14- and 15-year-old Florentine adolescent population.

## 2. Materials and Methods

### 2.1. Study Design and Population

This study has a cross-sectional design. The data collection took place between October 2021 and March 2022. In order to participate in this study, an invitation letter from the University of Florence was sent to the high schools of Florence. In particular, the availability of the first and second classes was requested, corresponding to 14 and 15 years. As a result, six schools have given their willingness to this study. The tests were carried out at the Asics Firenze Marathon Stadium “Luigi Ridolfi”. The inclusion criteria for the subjects were to not have any contraindications of physical activity and age within the range of ±6 months compared to the average age of their own class. Because not all adolescents participated in the physical fitness battery components, analysis groups and sample sizes vary for the different physical fitness tests.

The study was carried out in conformity with the ethical standards in the 1975 Declaration of Helsinki. This study is part of a project of the Tuscany Region and was approved in the Regional Prevention Plan 2014–2018 with the number O-Range18. All data were processed anonymously. Informed consent was obtained from legal guardians.

### 2.2. Physical Fitness Battery

The test battery thus included:Motor skills with t agility test;Cardio-respiratory endurance with 800 m run;Strength of lower limbs with vertical jump and long jump;Strength of upper limbs with handgrip and vortex launch;Flexibility with sit and reach;Sprint on 30 m with standing and launched start.

The t agility test includes running forward, sideways, and backward. Participants must face forward during the rehearsal and not cross their feet over each other. The time it took to complete the route was recorded in seconds [14]. 

An 800 m run assessed cardiorespiratory endurance. The test was performed in a standard sports field, and the students were asked to complete the distance as quickly as possible while maintaining a steady pace [15].

Lower limb strength was assessed by vertical and standing long jump. For the vertical jump test, the subjects stood flat-footed next to a wall with one arm extended above their head. The fingertips were stained with chalk powder, and they left their fingerprint on the wall in order to signal their standing reach height. After jumping as high as possible, they left a second chalk mark on the wall using the tips of their fingers, indicating the jumping reached height. Jumping height (expressed as cm) was determined by subtracting standing reach height from jumping reach height [16]. The long jump test assesses lower-limb explosive strength. After a short run, the subject jumps as far as possible, trying to land with both feet together. The length is obtained by measuring the distance in cm between the last heel stroke and the take-off line [15].

The handgrip test measures the maximal isometric strength generated mainly by the forearm. The dynamometer is adjusted to sex and hand size for each subject [17]. The vortex is a sports tool used in the youth categories of athletics as a primary throwing tool for the javelin throw. The vortex is an aerodynamic-shaped instrument consisting of a tail and a central body about 32 cm long, with about 30 cm in circumference, and weighing about 150 g, made of soft and light synthetic material. Higher scores (expressed in m) indicate better performance.

The sit-and-reach test measures the flexibility of the hamstring muscles. The test is performed with a standard box with a scale on the top. First, the participant must sit with the leg straight and the feet against the box and slowly reach forward as far as possible. Higher scores (expressed in cm) indicate better performance [18].

The 30 m sprint test measures the subject’s maximum acceleration capacity. These tests are performed by asking the subject to run as fast as possible after the starting signal in the case of a standing start. In the case of a launched start, 5 m is provided before the start, which allows for higher speed. The time was recorded in seconds [11].

### 2.3. Statistical Analysis

Data are shown as mean ± SD, percentiles from 3rd to 97th, and min–max values. The Shapiro–Wilk test was used to assess the normal distribution of data. Due to their asymmetric distribution, the Mann–Whitney U-test was used to compare continuous variables between groups. All calculations were carried out with IBM-SPSS^®®^ version 26.0 (IBM Corp., Armonk, NY, USA, 2019). A two-sided *p*-value < 0.05 was considered significant. 

## 3. Results

This study enrolled 1915 adolescents (mean age 15.0 ± 0.8 years; 54.0% females). In particular, 1028 belonged to the first class (mean age 14.3 ± 0.8 years; 54.5% females), and 887 belonged to the second class (mean age 15.3 ± 0.7 years; 53.4% females).

The results of the fitness tests show significant differences between the sexes in age groups and all assessments made (Table 1). In young male students, the comparison between the first and second classes shows an improvement in performance in all tests except in the sit-and-reach and the 30 m sprint-from-standing tests. In young female students, this comparison presents similar behavior, but in addition to the sit-and-reach test and the 30 m sprint with a standing start, the sprint with a launched start is also added (Table 1). 

Table 2, Table 3, Table 4 and Table 5 show the values from the third to the ninety-seventh percentile of the physical fitness tests. In detail, Table 2 reports the values of 14-year-old male students, Table 3 of 14-year-old female students, Table 4 of 15-year-old female students, and Table 5 of 15-year-old female students.

## 4. Discussion

This study analyzed the performance of 1915 subjects aged 14–15 years to generate, for the first time, sex- and age-specific normative values for physical fitness for Florentine adolescents. These normative values are added to the existing standards for the Florentine youth population on other cardiovascular risk factors related to physical exertion [19,20,21], adiposity [22], body mass index [23], eating habits [24], and lifestyle [25,26]. Although these fitness percentiles are not directly linked to the health status of the youth population, they help to monitor the phases of health surveillance. In addition, identifying the percentile rank of adolescents compared to their peers provides valid information for the early screening for any adverse trends. 

The results obtained from this study confirm that: the increase of the components of physical fitness during growth, even between the ages of 14–15, and the differences in physical performance between the sexes. Studies reporting normative physical fitness values for youth populations used heterogeneous test batteries [9,10,11]. Therefore, a direct comparison of values is only sometimes possible. However, handgrip and sit-and-reach tests are featured in numerous other studies [9,10,11]. In other studies, sprint tests were performed on 40 m [11], 50 m [15], or 60 m [10] instead of 30 m; 20 m shuttle run [9], or 600 m [10] instead of 800 m run was used for cardiorespiratory endurance. In addition, the t-agility test was proposed for college-age students [14].

Therefore, comparing the results obtained by the Florentine youth population with the reference values of the European standards [9] shows values superimposable of the strength of the upper limbs measured with the handgrip test, and values of the flexibility of the lower limbs worse in the subjects of this study. The t-agility test shows results comparable to those reported in the study by Pauole et al. [14]; considering the age differences, it is possible to hypothesize a consistent trend toward an increase. However, the 800 m test has values proportionally lower than the reference standards reported in studies of other European populations [10]. Furthermore, the strength of the lower limbs appears higher than the Spanish sports adolescents [27] peers in the vertical jump. This study’s best long jump values are the minimum values of the Italian peers who practice this discipline at a competitive level, matched by sex and age [28]. A result that could be expected is that the 30 m sprint shows better values with a start launched than a standing start. However, comparing these results over 30 m with those over 60 m by Blagus et al. [9] on young Slovenians of the same age, the best performances are comparable. At the same time, the reference values at lower speeds, therefore at the 97th percentile, are better in the present study. This could be attributable to fatigue in the second part of the sprint, from 30 to 60 m, carried out by Slovenian teenagers, especially in the unfit subjects. Finally, it is not possible to make a comparison with previous studies of the results obtained from the launch of the vortex. However, the authors opted for this tool used as a prerequisite for throwing the javelin because they believe that it is suitable for evaluation in any place, especially within the school, as a lightweight tool made of a rubbery material and increasingly known by the young population.

The main issues addressed in this study were recently investigated by a panel of international experts who defined the top 10 international priorities for research and surveillance on physical fitness among children and adolescents [29]. The present study confirms the guidelines suggested by the expert group, in particular, highlighting Priority 1 (Conduct Longitudinal Studies to Assess Changes in Fitness and Associations with Health), 2 (Use Fitness Surveillance to Inform Decision Making), and 7 (Assess the Reliability and Validity of Fitness Measures). About Priority 1: the percentiles elaborated in this study can be used to verify the changes in fitness status in response to an exercise program during growth; greater than expected changes for a given age, with an increase in the percentile value, could be attributed to the physical training. In particular, it could be used by physical education teachers as an initial assessment in the first high school class as proposed in Priority 4 (Implement Scalable School-Based Interventions to Improve and Promote Fitness). About Priority 2: national physical fitness surveillance systems can identify changes over the years. In Slovenia, for example, there has been a decrease in the performance of young people following the COVID-19 lockdown [30]; it would also be possible to verify the potential protective effects of physical activity interventions carried out during domestic confinement compared to those who did none [31]. Other countries use surveillance to identify areas with low fitness levels that therefore need intervention and later verify its effectiveness [32]. About Priority 7: as previously described, the present study used a different test battery from the other studies. At the same time, however, standardized test batteries are currently unavailable for physical fitness assessments. However, the authors believe that the assessments performed in the present study can be easily made in various settings. 

The physical fitness data collected in this study can contribute to Priority 5 (Develop Universal Health-Related Fitness Cut-Points) while leaving those sociocultural characteristics specific to the investigated area.

The present study has some limitations. Firstly, the battery of tests differs from those proposed in the European standards. However, as stated by the panel of international experts [29], there is no standard battery for physical fitness testing. The authors, therefore, chose the tests that best suited the sociocultural context of the Florentine area. In addition, the nine tests assessed all conditional abilities (upper and lower limb strength, cardiovascular endurance, speed, and flexibility)—an aspect the authors believe to be a strength of the present study. Secondly, the results of this study cannot be generalized to the entire Italian or European population, as the data was collected in a specific area. However, this is within the aim of the study, to provide specific data for a given territory with particular sociocultural characteristics. Thirdly, our normative values divided by sex and age could be affected by the different states of maturation between the different subjects. However, this aspect has yet to be evaluated in studies on this topic. Moreover, it would have added variables that compromised the possibility of elaborating normative values.

The study also has strengths. The first is that the subjects, being from the same territory, were evaluated with the same conditions, in the same place, and by the same operators. Therefore, the evaluation methodology was standardized. The second strength is the large sample size examined. This made it possible to provide percentile values with solid statistical values.

A future line of research is to lengthen the assessment age, covering both the pre-adolescent and late-adolescent phases. 

## 5. Conclusions

In summary, Florentine adolescents show lower values of muscular flexibility and cardiovascular endurance but greater strength and speed of the lower limbs than their European peers. The normative values of physical fitness reported in this study can be used to make initial assessments and to verify the effectiveness of training programs in adolescents. In addition, these data also guide policy decisions to achieve greater health for today’s young people who will be the adults of tomorrow. 

## Figures and Tables

**Table 1 healthcare-10-02486-t001:** Average values and st. dev. fitness tests divided according to sex and class categories. The differences between sexes in both the first and second class are always significant in any test with *p* < 0.001. * Significant differences between the first and second classes of the same sex.

	1 Class Males	1 Class Females	2 Class Males	2 Class Females
T-agility (s)	11.6 ± 1.3 *	12.9 ± 1.6 *	11.3 ± 1.55	12.6 ± 1.49
800 m (s)	338.8 ± 82.1 *	441.9 ± 89.9 *	315.3 ± 78.1	423.3 ± 85.4
Vertical jump (cm)	37.3 ± 9.2 *	29.5 ± 7.6 *	39.9 ± 9.7	31.7 ± 7.2
Long jump (cm)	364.4 ± 59.7 *	300.8 ± 51.5 *	396.6 ± 240.0	315.4 ± 50.4
Handgrip (kg)	35.4 ± 7.4 *	27.4 ± 4.7 *	38.5 ± 8.4	28.7 ± 4.7
Vortex (m)	29.7 ± 9.3 *	16.7 ± 5.3 *	32.7 ± 10.4	18.6 ± 5.7
Sit & reach (cm)	0.9 ± 8.1	9.3 ± 7.6	1.2 ± 9.4	9.6 ± 8.3
30 m S (s)	4.66 ± 0.62	5.22 ± 0.55	4.63 ± 0.65	5.21 ± 0.58
30 m L (s)	4.32 ± 0.54 *	4.88 ± 0.52	4.21 ± 0.51	4.85 ± 0.52

**Table 2 healthcare-10-02486-t002:** Fitness test percentiles for males in the first class of the high school.

		Percentiles								
	*n*.	3	5	10	25	50	75	90	95	97
T-agility (s)	310	9.7	9.9	10.2	10.7	11.4	12.3	13.2	13.9	14.2
800 m (s)	432	239	242	247	302	322	359	437	513	532
Vertical jump (cm)	454	21	23	26	31	37	43	50	51	55
Long jump (cm)	460	250	260	280	330	370	408	440	460	480
Handgrip (kg)	459	23	24	26	30	35	40	46	49	50
Vortex (m)	440	15	16	18	23	30	35	42	46	50
Sit and reach (cm)	437	−16	−14	−10	−4	2	6	11	13	15
30 m S (s)	157	3.7	3.7	3.9	4.2	4.6	5.0	5.5	5.7	5.9
30 m L (s)	401	3.7	3.7	3.8	4.0	4.2	4.5	4.9	5.2	5.6

**Table 3 healthcare-10-02486-t003:** Fitness test percentiles for females in the first class of the high school.

		Percentiles								
	*n*.	3	5	10	25	50	75	90	95	97
T-agility (s)	453	10.9	11.1	11.5	12.2	12.8	13.6	14.5	15.4	16.0
800 m (s)	480	319	327	339	356	430	509	550	617	633
Vertical jump (cm)	536	15	18	20	24	30	35	39	42	43
Long jump (cm)	548	200	210	230	270	300	340	370	386	400
Handgrip (kg)	537	20	20	22	24	27	30	34	35	37
Vortex (m)	524	9	10	11	13	15	20	24	28	30
Sit and reach (cm)	541	−8	−4	−1	5	10	14	18	21	22
30 m S (s)	228	4.3	4.4	4.6	4.8	5.2	5.5	5.9	6.2	6.6
30 m L (s)	494	4.1	4.2	4.3	4.5	4.8	5.1	5.5	5.8	6.0

**Table 4 healthcare-10-02486-t004:** Fitness test percentiles for males in the second class of the high school.

		Percentiles								
	*n*.	3	5	10	25	50	75	90	95	97
T-agility (s)	318	9.4	9.6	9.9	10.3	11.0	11.8	12.9	13.8	14.5
800 m (s)	380	230	234	240	251	312	341	417	443	519
Vertical jump (cm)	408	18	23	27	34	40	46	52	55	57
Long jump (cm)	402	250	260	290	340	390	430	460	489	505
Handgrip (kg)	411	23	25	28	33	38	43	49	52	55
Vortex (m)	388	16	17	20	25	32	40	46	51	56
Sit and reach (cm)	409	−17	−15	−13	−5	2	7	12	16	19
30 m S (s)	149	3.7	3.7	3.8	4.1	4.6	5.0	5.4	5.8	6.1
30 m L (s)	357	3.6	3.6	3.7	3.9	4.1	4.4	4.9	5.1	5.4

**Table 5 healthcare-10-02486-t005:** Fitness test percentiles for females in the second class of the high school.

		Percentiles								
	*n*.	3	5	10	25	50	75	90	95	97
T-agility (s)	399	10.4	10.7	11.2	11.7	12.5	13.4	14.4	15.17	15.4
800 m (s)	367	311	314	323	349	420	454	532	600	620
Vertical jump (cm)	462	18	20	22	27	32	37	41	43	45
Long jump (cm)	447	220	230	250	290	320	350	380	400	420
Handgrip (kg)	465	21	22	23	25	28	32	35	36	38
Vortex (m)	437	10	11	12	14	17	22	26	30	32
Sit and Reach (cm)	467	−8	−6	−1	5	11	15	19	22	23
30 m S (s)	175	4.1	4.2	4.5	4.8	5.2	5.6	6.0	6.2	6.4
30 m L (s)	414	4.1	4.2	4.3	4.5	4.7	5.1	5.5	5.9	6.1

## Data Availability

The data presented in this study are available on request from the corresponding author.

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
