# Peer review of "Florentine Normative Values for Physical Fitness in Adolescents Aged 14–15 Years"

_healthcare, 2022, doi:10.3390/healthcare10122486_

Round 1

Reviewer 1 Report

This is a valuable study that described the sex- and age-specific physical fitness reference values for a population of 14- and 15-year-old adolescents in Florence. Since there are no normative values for physical fitness for Italian youths, it is well understood that specific reference values for sex and age are needed to provide individualized interventions.

However, this is a very simple descriptive study that presents sex- and age-specific fitness values for 14- and 15-year-old adolescents in Florence. Therefore, I believe that it does not fully meet the criteria for the Original Research Articles due to the lack of novelty of the research question and its contribution to the advancement of science.

Author Response

Reviewer 1

This is a valuable study that described the sex- and age-specific physical fitness reference values for a population of 14- and 15-year-old adolescents in Florence. Since there are no normative values for physical fitness for Italian youths, it is well understood that specific reference values for sex and age are needed to provide individualized interventions.

However, this is a very simple descriptive study that presents sex- and age-specific fitness values for 14- and 15-year-old adolescents in Florence. Therefore, I believe that it does not fully meet the criteria for the Original Research Articles due to the lack of novelty of the research question and its contribution to the advancement of science.

The authors would like to thank the reviewer for appreciating our work. However, the second sentence's content could be clearer to the authors: the present manuscript seeks to fill a gap in normative references for fitness levels in the adolescent population. In the authors' view, this is the innovative part that contributes to the progress of science. In addition, this manuscript provides a tool that can be used by professionals in the field of sports and exercise science during their daily work practice. Numerous original research articles in the literature report percentile values in various research fields. Finally, the authors cannot provide specific answers as they have not received specific requests to improve their manuscript.

Reviewer 2 Report

Dear Authors,

Your paper is crystal clearly written and stands up for research among subpopulations in specific areas.

The only error in the paper is

·         You do not give a comprehensive graphical display of your result, as discussed in chapters 4, and 5. To remedy this, I recommend a polar plot, of Tables 2 – 5. One plot with nine ‘axes’ (the nine measurements in the tables and along the axes the percentile values). Colors should help the reader to distinguish your results and support in one glance your arguments in chapters 4 and 5. See below my item 7.

Below I also give recommendations to battle small errors:

Questions, suggestions, and discussions:

1.       Page 2, line 51,‘of the place’ does not generalize your work very well. You could generalize here by saying: ‘of subpopulations at places’.

2.       Page 2, lines 77, 80, and 81, display tests with capital characters. This is inconsistent with lines 100 and 107 where the handgrip and sit & reach do not have capitals s and r. Please make sure that your spelling is consistent everywhere, for instance, to confirm the first columns of Tables 2 – 5.

3.       Page 2, line 78, is the first occurrence with a blank before the meter: 800   m. It should be without blanks, see your lines 111, 156, and 171. Please make it consistent.

4.       Page 2, line 84, saying ‘T-test is an agility test’ is a wrong statement in your context of research. The T-test is an international acronym: https://en.wikipedia.org/wiki/Student%27s_t-test.  Please change it, e.g. “The T agility test includes running forward, . . .”.

5.       Page 2, lines 86, The second test ‘An 800 m run’ should begin at a new line.

6.       Page 3, the Table 1’s readability should be increased by deleting the two columns ‘Between sex’. The uniform content of these two columns should be said in the title capture.

7.       Pages 4, 5, the Tables 2 – 5, do need a visual summary. For instance in a polar chart with each test on a radius line, that is T-agility = East on the horizontal line, 800 m = North East on the 45 degrees line, and so on. This can be done in Excel or in Maple.

8.       Page 5, in lines 166, 167, and 169, you say ‘Furthermore, In addition, Furthermore’. This is too repetitive. I recommend replacing ‘In addition this’ simply with ‘This’. And I recommend replacing the second ‘Furthermore’ by the end of its sentence: ‘A result that could be expected is that the 30m sprint shows better …’.

Page 7, lines 238, 241, The double quotes should be removed.

Author Response

Reviewer 2

Dear Authors,

Your paper is crystal clearly written and stands up for research among subpopulations in specific areas.

The authors would like to thank the reviewer for appreciating our work and suggestions provided to improve our manuscript. Changes made to the manuscript are highlighted in red. Below are the answers to the reviewer’s comments. 

The only error in the paper is

  •         You do not give a comprehensive graphical display of your result, as discussed in chapters 4, and 5. To remedy this, I recommend a polar plot, of Tables 2 – 5. One plot with nine ‘axes’ (the nine measurements in the tables and along the axes the percentile values). Colors should help the reader to distinguish your results and support in one glance your arguments in chapters 4 and 5. See below my item 7.

Answer: Thank you for the suggestion. The authors have tried to work out the polar plot. However, the variables to be illustrated with percentiles along the radii have very different scales; therefore, they have very different values, and the scale of the polar diagram is only one. So variables that usually develop along the ray and others that all gather in a small space unless we have separate scales for each ray. Based on the trial we carried out, the authors did not denote any improvement in understanding the results obtained compared to the table, which appears to be a daily work tool for sports and exercise science professionals.

If the authors have misunderstood the reviewer's requests, they are available to provide the raw data to the reviewer for elaborating the requested graphs.

Below I also give recommendations to battle small errors:

Questions, suggestions, and discussions:

  1.       Page 2, line 51,‘of the place’ does not generalize your work very well. You could generalize here by saying: ‘of subpopulations at places’.

Answer: Thank you for the suggestion. The modification has been made accordingly.

  1.       Page 2, lines 77, 80, and 81, display tests with capital characters. This is inconsistent with lines 100 and 107 where the handgrip and sit & reach do not have capitals s and r. Please make sure that your spelling is consistent everywhere, for instance, to confirm the first columns of Tables 2 – 5.

Answer: Thank you for the suggestion. The authors agree with reviewer. The modification has been made accordingly.

  1.       Page 2, line 78, is the first occurrence with a blank before the meter: 800   m. It should be without blanks, see your lines 111, 156, and 171. Please make it consistent.

Answer: Thank you for the suggestion. The modification has been made accordingly.

  1.       Page 2, line 84, saying ‘T-test is an agility test’ is a wrong statement in your context of research. The T-test is an international acronym: https://en.wikipedia.org/wiki/Student%27s_t-test.  Please change it, e.g. “The T agility test includes running forward, . . .”.

Answer: Thank you for the suggestion. The modification has been made accordingly.

  1.       Page 2, lines 86, The second test ‘An 800 m run’ should begin at a new line.

Answer: Thank you for the suggestion. The modification has been made accordingly.

  1.       Page 3, the Table 1’s readability should be increased by deleting the two columns ‘Between sex’. The uniform content of these two columns should be said in the title capture.

Answer: Thank you for the suggestion. The modification has been made accordingly. Table 1 caption now is: “The differences between sexes in both first and second class are always significant in any test with p

  1.       Pages 4, 5, the Tables 2 – 5, do need a visual summary. For instance in a polar chart with each test on a radius line, that is T-agility = East on the horizontal line, 800 m = North East on the 45 degrees line, and so on. This can be done in Excel or in Maple.

 Answer: Thank you for the suggestion. The authors have tried to work out the polar plot. However, the variables to be illustrated with percentiles along the radii have very different scales; therefore, they have very different values, and the scale of the polar diagram is only one. So variables that usually develop along the ray and others that all gather in a small space unless we have separate scales for each ray. Based on the tests we carried out, the authors did not denote any improvement in understanding the results obtained compared to the table, which appears to be a daily work tool for sports and exercise science professionals.

If the authors have misunderstood the reviewer's requests, they are available to provide the raw data for elaborating the requested graphs.

  1.       Page 5, in lines 166, 167, and 169, you say ‘Furthermore, In addition, Furthermore’. This is too repetitive. I recommend replacing ‘In addition this’ simply with ‘This’. And I recommend replacing the second ‘Furthermore’ by the end of its sentence: ‘A result that could be expected is that the 30m sprint shows better …’.

Answer: Thank you for the suggestion. The modification has been made accordingly.

Page 7, lines 238, 241, The double quotes should be removed.

Answer: Thank you for the suggestion. The modification has been made accordingly.

Reviewer 3 Report

Dear Sirs,

Your paper seems well structured and of scientific interest.

The sample size is significant considering the specific age group You investigated.

Some aspects have to be addressed.

Please, indicate the study desing also in the title and in the abstract.

Line 16 (and in the rest of the paper): please don't start a sentence with a number.

In table 1 please remove comma within the cell: 18.6±5.,7

Lines 77-82: please create a bulleted list

The discussion section should be integrated in order to give Your results some clinical implications. For example, in which way these findings could help to improve prevention strategies for the common adolescence diseases? How these results could improve the dissemination of healthy lifestyles among children? To do that, I suggest the following references:

- Notarnicola, A., Farì, G., Maccagnano, G., Riondino, A., Covelli, I., Bianchi, F. P., . . . Moretti, B. (2019). Teenagers’ perceptions of their scoliotic curves. an observational study of comparison between sports people and non- sports people. Muscles, Ligaments and Tendons Journal, 9(2), 225-235.

-Farì, G., Di Paolo, S., Ungaro, D., Luperto, G., Farì, E., & Latino, F. (2021). The impact of covid-19 on sport and daily activities in an italian cohort of football school children. International Journal of Athletic Therapy and Training, 26(5), 274-278. 

-Bonavolontà, V., Cataldi, S., Maci, D., & Fischetti, F. (2020). Physical activities and enjoyment during the lockdown: Effect of home-based supervised training among children and adolescents. Journal of Human Sport and Exercise, 15(Proc4), 1338-1343. 

Please check that all references are reported as recommended by the ACS style guide.

Best regards and good luck

Author Response

Reviewer 3

Dear Sirs,

Your paper seems well structured and of scientific interest.

The sample size is significant considering the specific age group You investigated.

Some aspects have to be addressed.

The authors would like to thank the reviewer for appreciating our work and suggestions provided to improve our manuscript. Changes made to the manuscript are highlighted in red. Below are the answer to the reviewer’s comments. 

Please, indicate the study desing also in the title and in the abstract.

Answer: Thank you for the suggestion. The modification has been made accordingly. However, the authors would prefer to keep the more informative aspects of the content of the results and the research objective in the title rather than the study design, which, however, remains explicit to the reader already now from the abstract.

Line 16 (and in the rest of the paper): please don't start a sentence with a number.

Answer: Thank you for the suggestion. The modification has been made accordingly.

In table 1 please remove comma within the cell: 18.6±5.,7

Answer: Thank you for the suggestion. The authors were unaware of the typo

Lines 77-82: please create a bulleted list

Answer: Thank you for the suggestion. The modification has been made accordingly.

The discussion section should be integrated in order to give Your results some clinical implications. For example, in which way these findings could help to improve prevention strategies for the common adolescence diseases? How these results could improve the dissemination of healthy lifestyles among children? To do that, I suggest the following references:

- Notarnicola, A., Farì, G., Maccagnano, G., Riondino, A., Covelli, I., Bianchi, F. P., . . . Moretti, B. (2019). Teenagers’ perceptions of their scoliotic curves. an observational study of comparison between sports people and non- sports people. Muscles, Ligaments and Tendons Journal, 9(2), 225-235.

-Farì, G., Di Paolo, S., Ungaro, D., Luperto, G., Farì, E., & Latino, F. (2021). The impact of covid-19 on sport and daily activities in an italian cohort of football school children. International Journal of Athletic Therapy and Training, 26(5), 274-278. 

-Bonavolontà, V., Cataldi, S., Maci, D., & Fischetti, F. (2020). Physical activities and enjoyment during the lockdown: Effect of home-based supervised training among children and adolescents. Journal of Human Sport and Exercise, 15(Proc4), 1338-1343. 

Please check that all references are reported as recommended by the ACS style guide.

Answer: Thank you for the suggestion. In the authors' opinion, the only reference that could be used is Bonavolontà, V., Cataldi, S., Maci, D., & Fischetti, F. (2020). Physical activities and enjoyment during the lockdown: Effect of home-based supervised training among children and adolescents. Journal of Human Sport and Exercise, 15(Proc4), 1338-1343. Because scoliosis is not a topic covered in the manuscript, and the second article deals specifically with young athletes and soccer players. In contrast, this manuscript is aimed at both the sporting and non-sporting populations.

Therefore, a sentence addressing the topic of the pandemic was included in the discussion. Now the complete sentence is: 

“In Slovenia, for example, there has been a decrease in the performance of young people following the COVID-19 lockdown [30]; it would also be possible to verify the potential protective effects of physical activity interventions carried out during domestic confinement compared to those who did not do it [31].”

Best regards and good luck

Round 2

Reviewer 1 Report

Considering the importance and novelty of this research topic and its contribution to the development of science, it does not fully meet the criteria for an original research article.

Answer: The present manuscript seeks to fill a gap in normative references for fitness levels in the adolescent population. In the authors' view, this is the innovative part that contributes to the progress of science. In addition, this manuscript provides a tool that can be used by professionals in the field of sports and exercise science during their daily work practice. Numerous original research articles in the literature report percentile values in various research fields; therefore, it is not clear to the authors what criteria the reviewer refers to in order to define an original article. Finally, the authors cannot provide specific answers as they have not received specific requests to improve their manuscript